# Three-stage ultrafast demagnetization dynamics in a monolayer ferromagnet

Na Wu [1,2,4], Shengjie Zhang[1,2,4], Daqiang Chen [1,2], Yaxian Wang [1] ✉ & Sheng Meng [1,2,3] ✉

Intense laser pulses can be used to demagnetize a magnetic material on an extremely short timescale. While this ultrafast demagnetization offers the potential for new magneto-optical devices, it poses challenges in capturing coupled spin-electron and spin-lattice dynamics. In this article, we study the photoinduced ultrafast demagnetization of a prototype monolayer ferromagnet $Fe_3GeTe_2$ and resolve the three-stage demagnetization process characterized by an ultrafast and substantial demagnetization on a timescale of 100 fs, followed by light-induced coherent $A_{1g}$ phonon dynamics which is strongly coupled to the spin dynamics in the next 200–800 fs. In the third stage, chiral lattice vibrations driven by nonlinear phonon couplings, both in-plane and out-of-plane are produced, resulting in significant spin precession. Non-adiabatic effects are found to introduce considerable phonon hardening and suppress the spin-lattice couplings during demagnetization. Our results advance our understanding of dynamic charge-spin-lattice couplings in the ultrafast demagnetization and evidence angular momentum transfer between the phonon and spin degrees of freedom.

Energy dissipation caused by the flow of electrical current is a significant hurdle that hinders progress in nanoelectronics. One promising solution on the horizon is to encode data in the electron spin instead of charge, though with the grand challenge of controlling the static and dynamic behavior of spins and their ensembles. Meanwhile, spin dynamics itself is of fundamental interest as it unravels the complex interplay between spins, photons, charges, and phonons on ultrafast timescales (femtoseconds to picoseconds)[1–4]. The landmark observation of ultrafast demagnetization in nickel[5] demonstrates the potential to rapidly manipulate spins within a few hundred femtoseconds after excitation by an ultrafast laser pulse. Ever since then, tremendous efforts have been made to understand the light-driven transient magnetization dynamics[2,4,6–11]. The advent of the two dimensional (2D) magnetic materials, few(mono)-layers and their heterostructures, has further stimulated a flurry in the field of ultrafast magnetism and all-optical switch[10,12]. Among them, a widely studied example is two-dimensional ferromagnet $Fe_3GeTe_2$ (abbreviated

as FGT hereafter), embracing a variety of advantages including high and tunable Curie temperature[13–16], strong electron correlation[17], nontrivial topological band structure[18]etc. When this material is pumped by a femtosecond laser pulse, intriguing phenomena including photoinduced room-temperature ferromagnetism[19], spin transfer to proximity layers[20], two-step ultrafast demagnetization[21], as well as coherent phonon modulations[22], have been observed.

Nevertheless, despite these progresses the dominant interaction mechanism and microscopic origin of the ultrafast spin dynamics in two-dimensional magnets are not yet fully understood, especially when the coupling between electron spins and other degrees of freedom under *nonequilibrium* conditions are concerned. Specifically, seminal works highlight the crucial role of spin-lattice coupling, where the light-driven coherent lattice vibrations are well beyond the classical description of a quasi-equilibrium phonon bath, by demonstrating light-induced collective spin excitations, generation of chiral phonons, and light-induced magnetic phase transitions in systems including

[1]Beijing National Laboratory for Condensed Matter Physics and Institute of Physics, Chinese Academy of Sciences, Beijing 100190, China. [2]School of Physical Sciences, University of Chinese Academy of Sciences, Beijing 100190, China. [3]Songshan Lake Materials Laboratory, Dongguan, Guangdong 523808, China. [4]These authors contributed equally: Na Wu, Shengjie Zhang. ✉e-mail: yaxianw@iphy.ac.cn; smeng@iphy.ac.cn

$CoF_2$ and $ErFeO_3$[23,24]. Such strong spin-lattice coupling has also been shown in FGT by various signatures including the anomalous phonon linewidth[25,26], pressure-dependent spin configuration[27], and has been discussed in the aspect of strain-tunable magnetism proposed by equilibrium theory[28–30]. Therefore, a full understanding that describes the nonadiabatic effects under nonequilibrium conditions and captures the coupled electron-phonon-spin dynamics in the femtosecond time regime becomes urgently needed to calibrate and guide ultrafast demagnetization in low dimensional magnetic materials.

In this work, we present a full ab initio description of the ultrafast photoinduced demagnetization dynamics in the monolayer FGT, which features a *three-stage* process corresponding to a variety of microscopic interaction mechanisms, schematically summarized in Fig. 1. First, a rapid demagnetization originating from the electron degree of freedom takes place via spin-orbit coupling within a timescale of ~100 fs upon photoexcitation. Following this, there is a mild demagnetization in the next 600 fs which clearly shows an oscillating feature and is predominantly coupled with the out-of-plane $A_{1g}$ lattice vibration. Lastly, the in-plane phonon modes driven from nonlinear phononics grow in amplitude and can dominate the spin dynamics at a later stage, i.e. 800 fs after laser excitation. Interestingly, these in-plane lattice vibrations can result in a "tilting" of the magnetic moment towards the x-y plane, resembling the macroscopic picture of spin precession under an external magnetic field with a polar angle as large as ~20°. Such a significant spin precession with respect to the z-axis is a strong indication for angular momentum transfer to the chiral phonons arising from nonlinear phonon interactions. When driven coherently, such temporal evolution of magnetization can possibly evolve into collective excitations, known as spin waves or magnons. Further, we explicitly illustrate the pronounced nonadiabatic effect, which results in phonon hardening and suppresses the spin-phonon coupling. Our work provides a general picture for the photoinduced spin relaxation in monolayer ferromagnets via spin-orbit and spin-lattice interactions, therefore paving the way for finding efficient routes to manipulate various microscopic interactions to control macroscopic low-dimensional magnetism by light.

## Results

### The three-stage ultrafast demagnetization dynamics

Bulk $Fe_3GeTe_2$ crystallizes in a hexagonal structure and belongs to space group $P6_3/mmc$ (No. 194). Its monolayer has a sandwich structure with a mirror plane, consisting of five sublayers with two Te atoms occupying the top and bottom layers. The second and fourth layers contain two Fe atoms ($Fe_I$), while the middle layer is occupied by Ge and $Fe_{II}$ atoms (see Figs. 2a and 3a for the atomic structure). The optimized lattice constant is $a = 3.991$ Å, with a net magnetization of ~5.02 $\mu_B$/f.u. (f.u. denotes formula unit), both of which are in agreement with previous studies[30–32].

We employ the state-of-art real-time time-dependent density functional theory (rt-TDDFT) with Ehrenfest molecular dynamics to investigate the femtosecond-scale magnetization dynamics upon photoexcitation under nonequilibrium conditions. We optically excite the system using a Gaussian-enveloped linearly polarized laser with a photon energy of 1.5 eV and pump fluence of 1.93 mJ/cm² (see *Methods* for more details). To explicitly unravel the interactions between spin and other degrees of freedom including electron and lattice, we simulate the time-dependent magnetization dynamics including different interactions. More specifically, the lattice degree of freedom can be turned on and off by relaxing or fixing atomic positions along various axes, while the spin-orbit coupling (SOC) interactions can be blocked by ignoring the $H_{soc} \propto \vec{\sigma} \cdot \vec{L}$ term in the first-principle Hamiltonian. Thus, the magnetization evolution, plotted in Fig. 2b, is established under the following four scenarios: (1) without phonons or SOC (in light blue); (2) with phonons but without SOC (in cyan); (3) without phonons but with SOC (in yellow); (4) with both phonons and SOC (in brown). The comparison between the latter two cases allows one to examine the role of phonons or the lattice degree of freedom. Looking at the first two cases, it is obvious that there is no magnetic moment variation in the absence of SOC. These results are aligned with a seminal theoretical model[3], as well as a recent ab initio investigation[33] on the photoinduced demagnetization in nickel, confirming the necessity of non-spin-conserving interactions to modulate the spin of excited electrons.

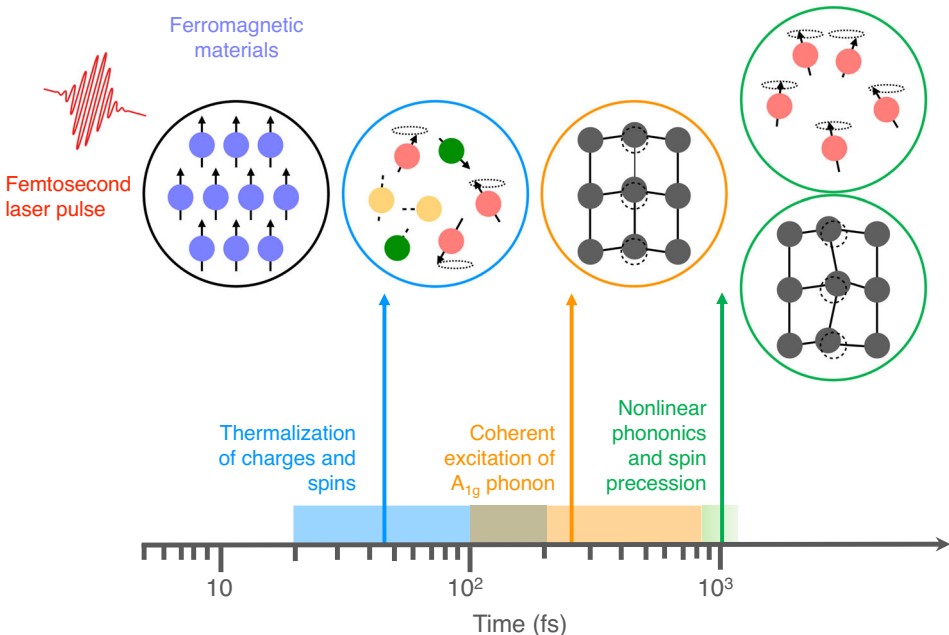

**Fig. 1 | The three-stage demagnetization mechanisms.** When a femtosecond laser pulse interacts with the charges and spins of a ferromagnet, the thermalization of the charges and spins occurs within 50–150 fs. After about 200 fs, a displacive excitation of coherent $A_{1g}$ modes is observed resulting from the photoexcitation-modulated potential energy surface. During the period of 800 fs to 1 ps, the nonlinear phonon coupling drives in-plane lattice motions which induce spin precession.

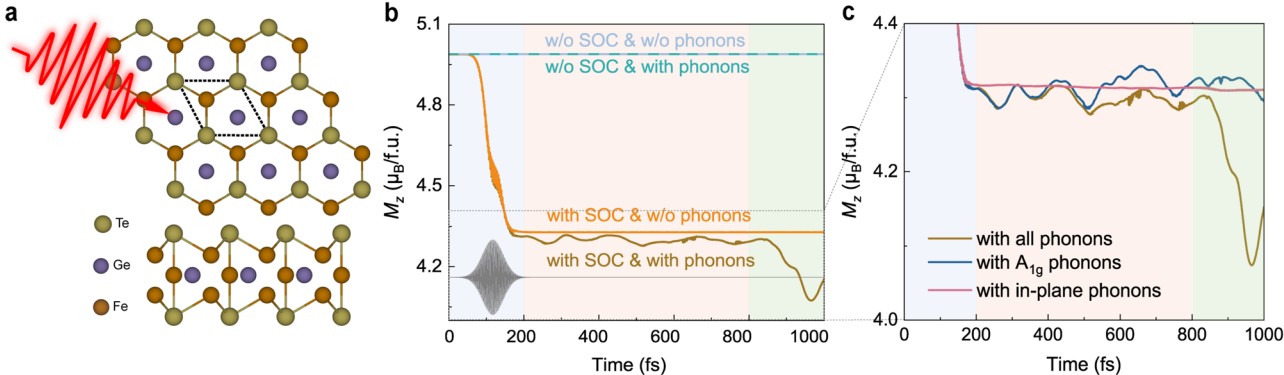

**Fig. 2 | Light-induced demagnetization dynamics in Fe₃GeTe₂. a** The top and side views of monolayer Fe₃GeTe₂ (FGT) crystal structure. The unit cell is labeled by the dashed line. **b, c** The three-stage photoinduced demagnetization in monolayer Fe₃GeTe₂ with color shading corresponding to various interaction mechanisms in Fig. 1. **b** The transient magnetization $M_z$ calculated under the four cases described in the text. The profile of the laser pulse is shown in grey and plotted in Fig. S1 in the

Supplemental Information (SI). **c** Enlarged presentation of stage II and III to highlight the contributions from different degrees of freedom for lattice vibrations: only coherent $A_{1g}$ modes (dark blue), only in-plane phonons (pink), and full lattice freedom corresponding to the same brown curve in **b**. The further change of $M_z$ after 800 fs clearly arises from a cooperative contribution of both in-plane and out-of-plane lattice vibrations.

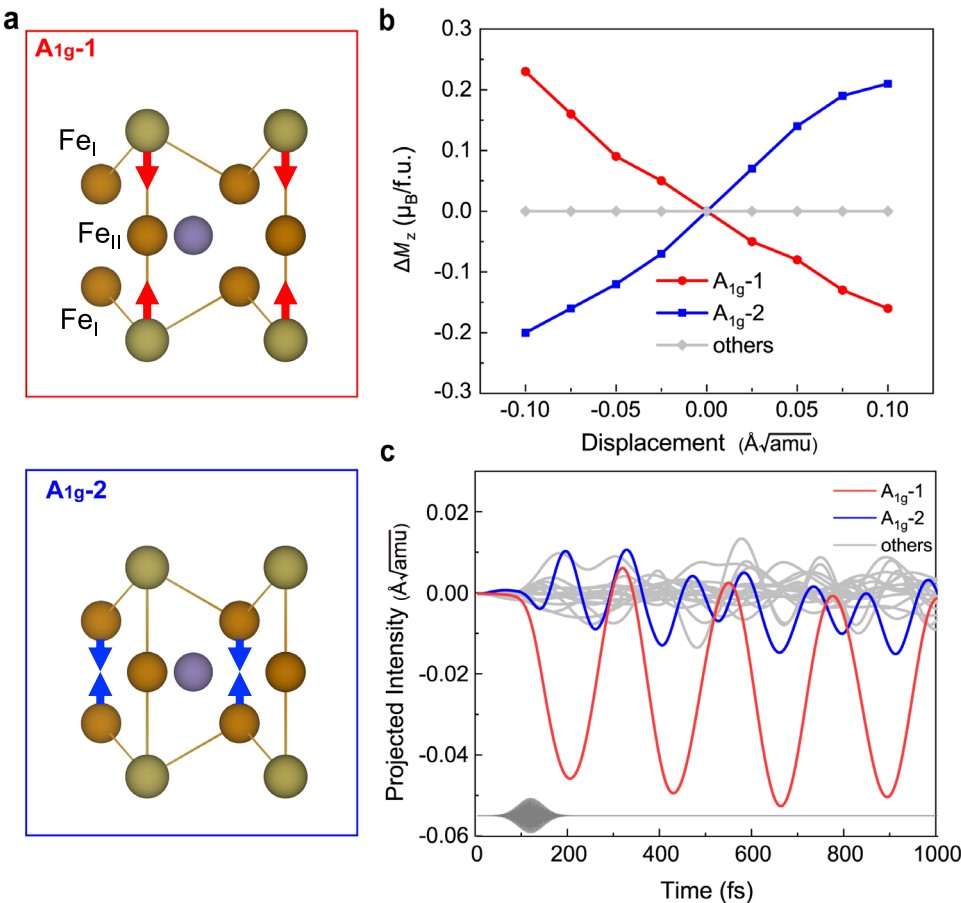

**Fig. 3 | The static and dynamic spin-phonon coupling. a** Atomic motion corresponding to the two $A_{1g}$ phonon modes strongly coupled with $M_z$, with the arrows designated as "positive" displacement, as in **b**. **b** The variation of the $z$-component magnetic moment $\Delta M_z$ upon lattice distortion along the phonon eigenvectors with

amplitude ranging from $-0.10\ \text{Å}\sqrt{\text{amu}}$ to $0.10\ \text{Å}\sqrt{\text{amu}}$. The red (blue) line represents the case of $A_{1g}$-1 ($A_{1g}$-2) phonon and the grey represents the others. **c** Coherent excitation of the two $A_{1g}$ phonon modes in FGT upon photoexcitation by an ultrashort laser pulse with a photon energy of 1.5 eV and fluence of 1.93 mJ/cm².

With the presence of SOC, we immediately observe a rapid and substantial demagnetization on a timescale of ~100 fs within the duration of the pump pulse, corresponding to a fast spin and charge thermalization (blue block in Fig. 1). A substantial magnetic moment reduction of $0.60\ \mu_B$ per unit cell, or approximately $0.20\ \mu_B$ per Fe

atom is achieved, with no dependence on phonons (yellow and brown lines in Fig. 2b). The demagnetization happens mostly on the two $Fe_I$ atoms, with the Ge and Te atoms maintaining a zero magnetization during the process (for details see Fig. S7 in SI). We note that the demagnetization in FGT is over an order of magnitude greater than

that observed in bulk Ni (after-pulse demagnetization of roughly 0.06 $\mu_B$ per Ni atom[33]). Second, after the demagnetization from the electron bath, we see an oscillatory evolution during 200–800 fs, indicating that coherent phonons might play a role in the variation of $M_z$. Specifically, without lattice vibrations, the demagnetization of the system reaches a quasi-equilibrium state after the laser pulse disappears. However, when the lattice dynamics is enabled, there is a successive after-pulse demagnetization, an extra - 0.08 $\mu_B$/f.u., with an oscillatory feature taking place on a timescale of a few hundred femtoseconds (yellow block on the timeline in Fig. 1). We perform a fast Fourier transform (FFT) analysis on the $\Delta M_z$ dynamics from 200 to 800 fs and observe two coherent oscillation peaks at approximately 4.72 and 8.89 THz (for details see SI Section S4). These two frequencies correspond to the two $A_{1g}$ phonon modes, strongly indicating that the lattice degree of freedom plays an important role in the femtosecond magnetization dynamics which we will discuss in detail in the next session.

Even more surprisingly, there is another large decrease of $M_z$ of approximately 0.30 $\mu_B$/f.u. after 800 fs, which we attribute to the contribution of in-plane lattice vibrations (green block on the timeline in Fig. 1) for the following reasons. It becomes clear from Fig. 2c that $M_z$ keeps the oscillatory feature but shows no further decrease occurring after 800 fs with the presence of $A_{1g}$ phonons only (dark blue line in Fig. 2c). Meanwhile, with the presence of only in-plane atomic motion, there is negligible demagnetization on top of the electronic contribution (pink line in Fig. 2c), indicating that in-plane phonon modes by themselves have weak couplings with electronic spins. The rapid change in $M_z$ at the third stage, i.e. after 800 fs, only appears when all vibrational degrees are activated (brown line in Fig. 2c). This strongly indicates that stage III is governed by a different mechanism involving nonlinear or higher-order interactions.

### The spin-lattice coupling

To reveal the role of lattice degree of freedom in the process of demagnetization, we examine the mode-resolved spin-phonon coupling strength. Fig. 3b shows the variation of the $z$-component of the magnetic moment ($\Delta M_z$) when the atomic structure is distorted along each phonon eigenvector with an amplitude from $-0.10$ Å$\sqrt{\text{amu}}$ to $0.10$ Å$\sqrt{\text{amu}}$. One sees that $M_z$ is strongly subject to the two $A_{1g}$ phonon modes involving the atomic motion along the $z$-axis that retain the mirror and three-fold rotational symmetry, illustrated in Fig. 3a. More specifically, for the low frequency $A_{1g}$-1 optical mode with $\omega = 4.58$ THz, the positive (negative) displacement of atoms represents the direction of the decrease (increase) in the Te-Fe$_I$ bond length, while for the higher frequency $A_{1g}$-2 phonon with $\omega = 8.87$ THz, the positive (negative) displacement of atoms represents the direction of the increase (decrease) in Te-Fe$_I$ bond length, indicated by the arrows in Fig. 3a. For both modes, the increase of bond length between the Te and Fe$_I$ atoms leads to a positive variation of the $z$-component magnetic moment, and vice versa. We observe zero changes in $M_z$ upon distortion along the other phonon modes, shown by the grey lines in Fig. 3b, neither changes of the $x$- nor $y$-component of the magnetic moment.

The strength of the spin-phonon coupling (SPC) interactions can be quantified by a first-order parameter $\lambda_{SPC}$ as

$$\lambda_{SPC} = \left| \frac{\Delta M_z}{\Delta u} \right|, \tag{1}$$

where $\Delta M_z$ and $\Delta u$ refer to the variation in $M_z$ and the atomic displacement, respectively. The coupling strength for the two $A_{1g}$ modes is roughly 1.88 and 2.00 $\mu_B$/ (Å$\sqrt{\text{amu}}$), respectively, whereas those of all other phonon modes are precisely zero. We note that, the ferromagnetic behaviors in FGT have a strong dependence on its layer number[13,15,16], and the ultrafast dynamics can be quite anisotropic in

low-dimensional materials[34,35]. As the ferromagnetic order is more stable in the bulk sample, the spin-lattice coupling, i.e. the variation of $M_z$ upon lattice displacement is greatly enhanced in the monolayer system (for detailed discussion see SI Section S3).

The strong and highly selective spin-phonon coupling indicates a coherent phonon excitation during the ultrafast demagnetization. Here, the light-induced coherent lattice motion can be characterized by projecting the transient atomic displacements onto each phonon eigenvector, as illustrated in Fig. 3c. We do observe coherent excitation for the two $A_{1g}$ phonon modes, both with a -5% hardening compared to their ground state frequencies. The intensity of $A_{1g}$-1 mode, which mainly involves the displacement of Te atoms, is significantly greater than that of other modes, yet the excitation of $A_{1g}$-2 mode is also clearly visible and coherent. Recently, femtosecond transient optical spectroscopy measurements have also revealed the coherent $A_{1g}$ phonon excitation in FGT, which can be maximized with a 1.574 eV laser pump[31], verifying our simulations and analysis regarding the crucial role of $A_{1g}$ phonons in the ultrafast demagnetization dynamics. However, we do notice that the static calculations predict a large spin-phonon coupling parameter while the magnetization variation in the nonadiabatic TDDFT simulations rather oscillates without a substantial change during 200–800 fs, which we shall return to shortly.

### Nonlinear phononics and spin precession

Now we come to the puzzling spin dynamics after 800 fs, where an extra 30% demagnetization occurs, which can not be accounted for by the $A_{1g}$ phonons only (Fig. 2c). This is counter-intuitive given the fact that both the static and the molecular dynamics calculations show coupling between magnetization and mostly, if not only, the out-of-plane $A_{1g}$ phonons. It thus indicates that the in-plane lattice vibrations interact with the spin degree of freedom in this stage. Therefore, we now examine both the $x$- and $y$-component of the transient magnetization ($M_x$ and $M_y$) in addition to $M_z$, as well as the amplitude of $M_{tot} = \sqrt{M_x^2 + M_y^2 + M_z^2}$, as shown in Fig. 4b. In this regard, the demagnetization is dominated by the $z$-component until 800 fs, with both the $x$- and $y$- components remaining constantly at zero, while after 800 fs we see a further demagnetization of $M_z$ but an increasing magnetic moment in the in-plane direction, with $M_{tot}$ unchanged. This demonstrates a *transverse* spin relaxation, i.e. a rotation of the magnetic moment without relaxation in its amplitude. Such kind of dynamics is better illustrated when we plot the effective magnetic moment vector **m** on a sphere, shown in Fig. 4a, with colormap from blue to red corresponding to the temporal evolution in the first picosecond after photoexcitation. This highly resembles the classical spin precession of a magnetic moment under an external field, with the presence of a damping torque. The effective precession angle in our simulation reaches as large as -20°, indicating a considerable angular momentum transfer.

The transverse spin relaxation evidences that in-plane phonon modes, likely driven by the anharmonic coupling with the $A_{1g}$ modes, known as nonlinear phononics, start to significantly affect the magnetization dynamics after 800 fs. A complete presentation of the time evolution of all in-plane phonon modes can be found in SI, Section S5. Two doubly-degenerate infrared (IR) active modes are activated after 300 fs with their amplitudes increasing over time. Their superposition can account for excitation of the chiral phonon modes (Fig. S9 in SI), which in turn dominate the spin dynamics after 800 fs. To better visualize such an effect, the trajectory of Fe$_I$ atom in the $x$-$y$ planes is shown in Fig. 4d, with color coding denoting the temporal evolution again. The dashed circle indicates the phonon amplitude, i.e. the maximum atomic displacement in this case to guide the eye. The Fe$_I$ atom indeed exhibits a chiral motion in the $x$-$y$ plane, although with an elliptical trajectory rather than a well-defined circle. This might be due

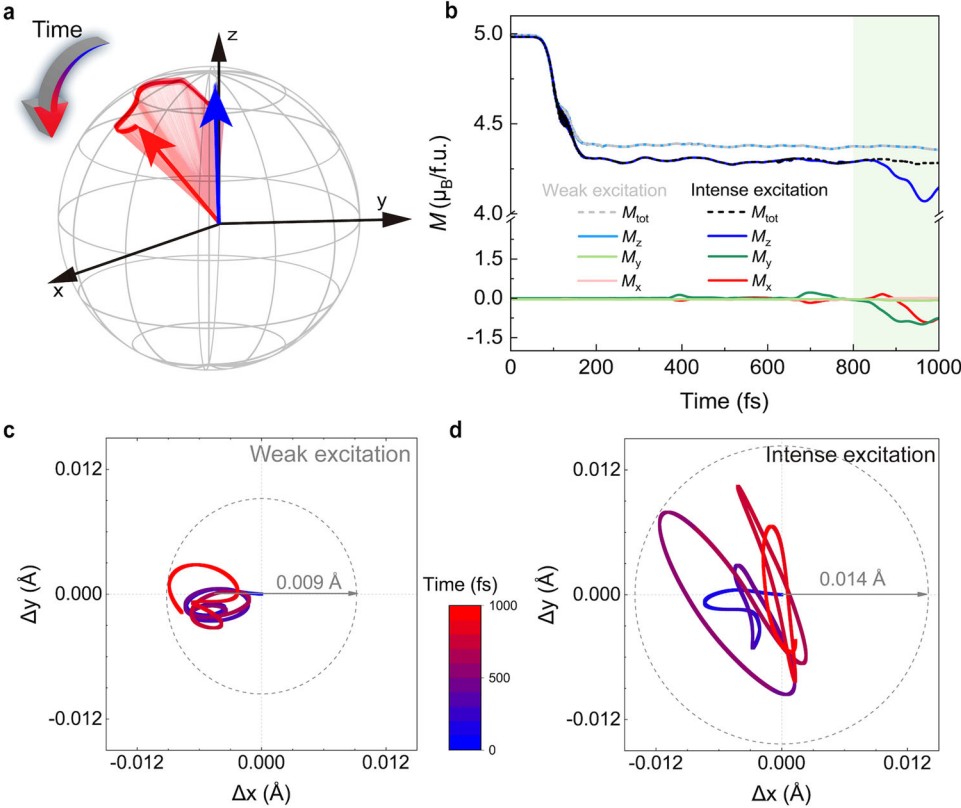

**Fig. 4 | The chiral phonon generation and spin precession. a** Real-time evolution of the magnetization, depicted as an effective magnetic moment **m** precession with respect to an external field along the z-axis. The arrow representing the **m** is color-coded by the temporal evolution, with the direction and magnitude from our real-time first-principle calculations. **b** The temporal evolution of total and the x-, y-, and z-components of the transient magnetization for the weak (1.71 mJ/cm²) and intense (1.93 mJ/cm²) excitation intensities, both with the photon energy of 1.5 eV. **c, d** Trajectories of $Fe_I$ atom in the x-y plane under weak and intense excitation, respectively. The dashed circles are drawn to display the magnitude of in-plane phonons to guide the eye, showing a much smaller atomic displacement (-0.009 Å) under weak excitation than that (-0.014 Å) under intense excitation. The colormap again denotes the simulation time from 0 to 1000 fs.

to that the chiral phonons driven from nonlinear phononics do not maintain a perfect coherence and the phase is strongly coupled with other phonon modes. Still, these results pinpoint that the spin dynamics in the later stage are governed by a different mechanism, highly involving chiral phonons generated from nonlinear phonon interactions. More importantly, although in the present case, the phonons tend to thermalize after 1000 fs (Secs. S5 and S6 in SI), which might suppress the spin precession, it is possible to optically drive the chiral phonons selectively. Once the phonon modes are coherently driven, they can maintain a better coherence, thus it is possible for a collective excitation of electron spins to display coherent precession in the form of a spin wave, i.e. magnons. A similar mechanism has been shown in the previous work on the effective magnetic field induced by optically driven phonons[24].

Moreover, the optical selectivity inspires us to evaluate the tunability of such an effect by the pump light intensity. As indicated by the lighter-colored lines in Fig. 4b, when we apply weak excitation with pump fluence of 1.71 mJ/cm², there is no transverse relaxation of the magnetic moment, as shown by the nearly-zero $M_x$ and $M_y$ in the entire time span. As discussed in details in SI, Section S9, under weak excitation, the optically driven $A_{1g}$ modes have much smaller amplitudes, leading to the absence of the nonlinear-phonon-induced infrared in-plane $E_{2u}$ modes, a superposition of which can generate chiral phonons. Indeed, the amplitude of the chiral atomic motion is significantly smaller when tracing the motion of $Fe_I$ atom in the x-y plane, shown in Fig. 4c. This provides further evidence that the photoinduced spin precession arises from angular momentum transfer to chiral phonons generated via nonlinear phonon interactions between the $A_{1g}$ and in-plane modes.

## The nonadiabatic effects

As mentioned previously, the magnitude of magnetization oscillations during the second stage (200–800 fs) is much smaller than the variation of the magnetic moment caused by the same phonon amplitude extrapolated from the static calculations. This is likely due to the nonadiabatic effects when the pump laser drives the system far away from equilibrium. The nonadiabatic effect has been shown to be non-negligible in metals and semimetals, as when the phononic and electronic energy scales become comparable, their motion cannot be adiabatically decoupled, thus breaking down the Born-Oppenheimer approximation[36–39].

To evaluate the significance of the nonadiabatic effects in FGT and their influence on the coupled spin-phonon dynamics, we perform Born-Oppenheimer molecular dynamics (referred to as BOMD) simulations to eliminate the nonadiabatic effect as a comparison. Specifically, we start from a distorted lattice structure with the two coherent $A_{1g}$ phonon modes initiated at $t = 0$, without the optical excitation so the electron occupation number evolves from the ground state. Figure 5a and c illustrate the time-dependent magnetization dynamics under both the adiabatic (BOMD, in lighter color) and the nonadiabatic (NAMD, in darker color) simulations, with coherent excitation of the $A_{1g}$-1 and $A_{1g}$-2 phonon, respectively. The bottom panels of Fig. 5a and c highlight the nonadiabatic effect, which manifests in a shortened period of both $A_{1g}$ phonons, i.e. a phonon hardening effect. For the ultrafast magnetization dynamics, we plot the magnitude of the magnetic moment variation, $|\Delta M_z| = |(M_z)_{max} - (M_z)_{min}|$, in the upper panels of Fig. 5a and c. For both $A_{1g}$ modes, the magnitude of magnetization oscillations is greatly reduced by more than 75% when the

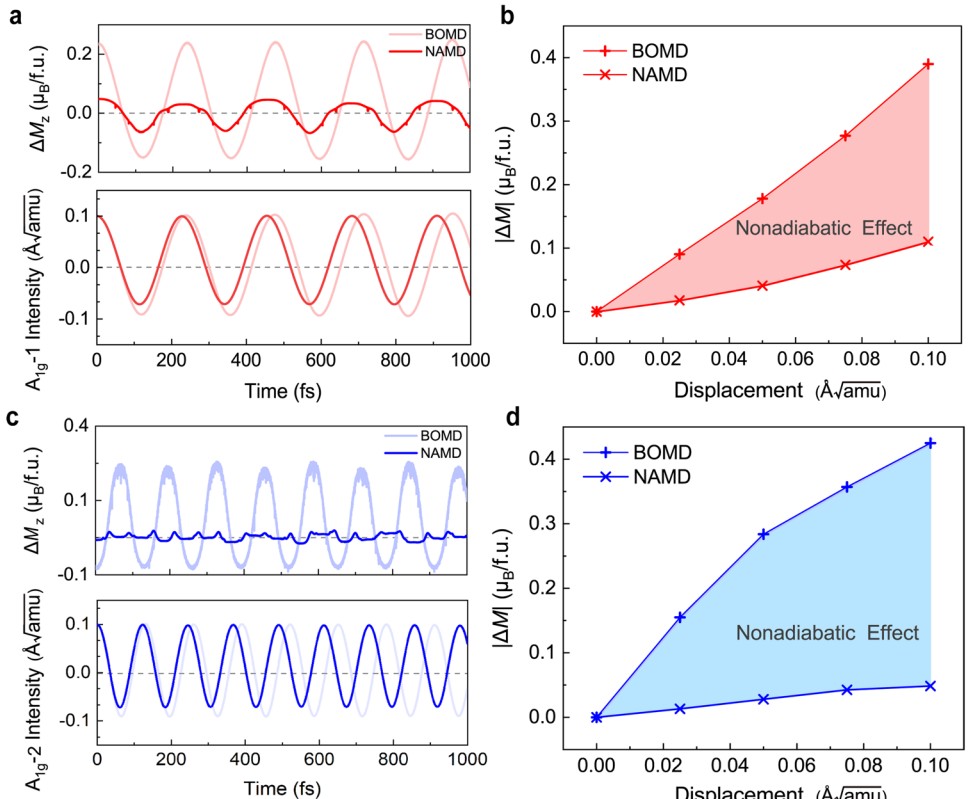

**Fig. 5 | The nonadiabatic effect. a**, **c** Variation of the $z$-component of transient magnetic moment $\Delta M_z$ and the intensity of the corresponding phonon mode obtained by the adiabatic and non-adiabatic simulations with an initial phonon amplitude of 0.10 $\text{Å}\sqrt{\text{amu}}$ of the two $A_{1g}$ coherent modes. **b**, **d** The relative change in $z$-component of magnetic moment with dependence on the phonon amplitude, with the shaded region representing the difference caused by the nonadiabatic effect. The symbols of plus (X) denote the results of BOMD (NAMD) simulations, respectively.

nonadiabatic effect is taken into account. Meanwhile, when the electron occupation number stays unchanged from the ground state, we did not observe any demagnetization, implying that the nonequilibrium excited carrier distribution will greatly affect the spin-phonon coupling (for details see SI Section S7).

Further, Fig. 5b and d shows the magnitude of magnetization oscillations with respect to various phonon amplitudes, with the shaded region emphasizing the contribution of the nonadiabatic effects. It again demonstrates that the nonadiabatic effect suppresses the spin-phonon coupling significantly, and the suppression increases with phonon amplitude. Using the same definition of the spin-phonon interactions as in Eq. (1), we note that the strength of spin-phonon coupling under adiabatic condition $\lambda_{\text{BOMD}}$ is significantly greater than that under nonadiabatic condition $\lambda_{\text{NAMD}}$ for both $A_{1g}$ modes. This demonstrates, for the first time to the best of our knowledge, that the nonadiabatic effect is non-negligible in excited state magnetization dynamics, and highlights the necessity that the electronic and phononic degrees of freedom have to be treated on an equal footing.

## Discussion

It is commonly accepted that ultrafast demagnetization can be characterized by stages with distinct characteristic timescales corresponding to different dominant relaxation processes, which can be embedded in the transient signals from various pump-probe measurements[3–5,33]. However, our results clearly demonstrate the predominant role of $A_{1g}$ coherent phonons and nonlinear phonon interactions at different demagnetization stages. This highlights a substantial difference between such a monolayer ferromagnet and the elementary ferromagnets such as Ni and Fe, where the phonon phase space is rather limited and the lattice mostly serves as a thermal bath.

Further, the large amplitude of the in-plane atomic motion (Fig. 4d) evidences the formation of chiral phonons, where the circular motions of atoms carry an intrinsic angular momentum. It was proposed that chiral phonons can generate a giant effective magnetic field[40], and here the magnetization precession dynamics may naturally serve as a unique signature for angular momentum transfer. From this perspective, our work paves the way for understanding the mechanism underlying novel phenomena related to electron-phonon-magnon coupling interactions, though the exact coupling between chiral phonons and coherent spin waves needs further investigation, both theoretically and experimentally. In addition, similar to the ultrafast demagnetization experiment of Ni, light-induced demagnetization can be potentially applied for terahertz emission[41–46], suggesting that FGT can be a promising terahertz emitter. The rich physics in the coupled spin-phonon dynamics also offers an alternative avenue for ultrafast, low-dissipation, and "all-optical" control of magnetic order.

In conclusion, we resolve the three-stage photoinduced ultrafast demagnetization dynamics in a two-dimensional ferromagnetic metal $Fe_3GeTe_2$, governed by distinct interaction mechanisms between various degrees of freedom. While a rapid demagnetization happens on a timescale of sub-hundred femtoseconds, the transient magnetic moment from 200 to 800 fs shows a clear coupling with photoexcited coherent $A_{1g}$ phonon modes. Further, we observe that nonlinear phonon interactions between the light-driven $A_{1g}$ modes and the in-plane phonons will introduce chiral phonon modes with an elliptical atomic trajectory, in accompany with a transverse spin relaxation, namely an effective magnetization precession after 800 fs. We anticipate that under delicate driving conditions, the coherent phonon modes can have an explicit coupling with collective spin excitations and may drive the formation of magnons. Further, we explicitly

address the nonadiabatic effect, which shortens the phonon period and considerably suppresses the spin-phonon coupling interaction, highlighting its roles under nonequilibrium conditions. Our work provides atomistic insights into ultrafast demagnetization dynamics of two-dimensional ferromagnets in the femtosecond timescale and emphasizes the crucial role of the lattice degrees of freedom. It further facilitates new device concepts in the quest for manipulating magnetism in an ultrafast and low-dissipation manner.

## Methods

### Theory

The excited state dynamics can be simulated via a real-time density functional theory framework (rt-TDDFT)[47–49], which we implemented in the QUANTUM ESPRESSO package[50–54]. More details on the theoretical framework and implementation are provided in SI Secs. S1 and S2. Under the theorem of TDDFT, the time evolution of the two-component Kohn-Sham spinors $\psi_{\gamma,\boldsymbol{k}}(\boldsymbol{r},t)$ follows the time-dependent Kohn-Sham equation[33,55]:

$$i\hbar \frac{\partial}{\partial t} \psi_{\gamma,\boldsymbol{k}}(\boldsymbol{r},t) = \hat{H}_{KS} \psi_{\gamma,\boldsymbol{k}}(\boldsymbol{r},t). \tag{2}$$

The Hamiltonian $\hat{H}_{KS}$ can be written as

$$\hat{H}_{KS} = -\sum \frac{\hbar^2}{2m_i} \nabla_i^2 + V_{ION} + V_H + V_{XC} + U_{ext} + V_{soc}, \tag{3}$$

where the terms in the equation represent, in order, the kinetic energy term of the electron $-\sum \frac{\hbar^2}{2m_i} \nabla_i^2$, the Coulomb potential generated by the nuclei $V_{ION}$, the Hartree potential $V_H$, the exchange-correlation potential $V_{XC}$, the external field $U_{ext}$ and the spin-orbit coupling term $V_{soc}$.

In the Heisenberg picture, the kinetic velocity is $\boldsymbol{v} = \frac{1}{i\hbar}[\boldsymbol{r},H] = \frac{1}{m}\left[\boldsymbol{p} + \frac{e}{c}(\boldsymbol{A}+\mathcal{A})\right]$, with $V = V_{ION} + V_H + V_{XC} + U_{ext} + V_{soc}$ denoting the potential generated by the ions, which acts on the electrons, and $\boldsymbol{\sigma}$ the Pauli matrices. Here $\mathcal{A} = \frac{\hbar}{4mce}\boldsymbol{\sigma} \times \nabla V$, coming from the contribution of SOC. This indicates that $\mathcal{A}$ can be seen as an effective magnetic field ($\mathcal{B} = \nabla \times \mathcal{A} = \hbar\left[\boldsymbol{\sigma}\cdot(\nabla^2 V) - (\boldsymbol{\sigma}\cdot\nabla)\nabla V\right]/4mc$) that affects the electron and spin dynamics due to the presence of the SOC.

The ion dynamics $\boldsymbol{R}_\alpha(t)$ follows the Hellmann-Feynman theorem[56]:

$$M_\alpha \frac{d^2}{dt^2} \boldsymbol{R}_\alpha(t) = -\sum_\gamma f_\gamma \langle \psi_\gamma | \hat{H}_{KS} | \psi_\gamma \rangle, \tag{4}$$

where $f_\gamma$ represents the occupation number of time-dependent Kohn-Sham wavefunctions. $M_\alpha$ and $\boldsymbol{R}_\alpha$ are the mass and position of the $\alpha$th ion.

By solving the coupled electron-ion motion (Eq. (3) and Eq. (4)), we can obtain the time-dependent total magnetization

$$M(t) = \int \boldsymbol{m}(\boldsymbol{r},t)\,d\boldsymbol{r}, \tag{5}$$

where the time-dependent local magnetization is

$$\boldsymbol{m}(\boldsymbol{r},t) = \mu_B \sum_{\gamma,\boldsymbol{k}} \psi_{\gamma,\boldsymbol{k}}^\dagger(\boldsymbol{r},t) \cdot \boldsymbol{\sigma} \cdot \psi_{\gamma,\boldsymbol{k}}(\boldsymbol{r},t), \tag{6}$$

with $\mu_B$ representing the Bohr magneton ($\mu_B = \frac{e\hbar}{2m}$)[33,57].

### Computational details

The structural relaxation and the electronic structure of monolayer FGT are calculated based on the density functional theory implemented in the QUANTUM ESPRESSO package[50–52], and the phonon frequency and eigenvectors are calculated within the density functional perturbation theory (DFPT)[58]. We employ the local density approximation (LDA) to describe the electronic exchange-correlation contribution to the total energy[59,60]. The valence electron wave functions are expanded as plane-wave basis sets with the energy cutoff of 120 Ry. We use the full-relativistic, norm-conserving pseudopotentials (NCPP) to describe core electrons and the nuclei[61,62]. The Brillouin zone is sampled by an $11 \times 11 \times 1$ Gamma-centered $k$-mesh. The structure and atomic positions are fully relaxed with a convergence threshold on the ionic forces and total energy being $10^{-9}$ a.u. and $10^{-10}$ Ry, respectively.

We apply a Gaussian-enveloped laser pulse following a waveform

$$\boldsymbol{E}(t) = \boldsymbol{E_0}\cos(2\pi\omega t)\exp\left[-(t-t_0)^2/2\sigma^2\right], \tag{7}$$

where the pulse duration $\sigma$ is 27.6 fs and the photon energy $\hbar\omega$ is 1.5 eV. The laser field reaches the maximum strength of $E_0 = 0.07$ V/Å at time $t_0 = 50$ fs. The polarized laser pulse has an electric field profile shown in Fig. S1 in the SI.

We employed a reduced $5 \times 5 \times 1$ $k$-mesh, and the time step is 0.145 femtosecond for nuclei and 0.145 attosecond for electrons in our dynamical simulations. The difference in their evolution timesteps comes from the implementation of the evolution operator, discussed in details in SI Section S2.

## Data availability

Relevant data supporting the key findings of this study are available within the article and the Supplementary Information file. All raw data generated during the current study are available from the corresponding authors upon request.

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

## Acknowledgements

The authors thank Dr. Chao Lian for helpful discussions. This work is supported by National Natural Science Foundation of China (Nos. 12025407, 11934003, and 92250303), Chinese Academy of Sciences (Nos. YSBR-047 and XDB33030100), and Ministry of Science and Technology (No. 2021YFA1400201).

## Author contributions

Y.W. and S.M. conceived the project and designed the research; N.W., S.Z., and D.C. performed the first principles calculations and data analysis; N.W. and D.C. modified and further developed the Magnetization Module in the time-dependent ab initio package with plane waves (TDAPw). N.W., Y.W., and S.M. wrote the paper. All authors discussed the results and contributed to the manuscript.

## Competing interests

The authors declare no competing interests.
