## [Peer Review File · Nature Communications]

Reviewers' Comments:

Reviewer #1:

Remarks to the Author:

The work by Wu et al. presents a theoretical investigation of the ultrafast dynamics of Fe₃GeTe₂ (FGT) using rt-TD-DFT technique. The paper is well-written and organized. Beyond the similar ultrafast demagnetization within 100 fs as previously investigated (Ref.[30]), there are very interesting observations about spin-lattice coupling from 100 fs to 1000 fs, including the selective coupling of the A_{1g} phonons to magnetism, and additional demagnetization after 800 fs. However, some of the main conclusions are not yet well supported due to the outstanding questions listed below. In addition, it is not clear if these results are uniquely related to the monolayer structure, or apply to bulk FGT, even 3D magnets. Without these discussions, the impact of the work could be limited. In summary, the manuscript has great potential but needs a major revision to meet the criteria for publishing on Nature Communications.

1. The title mentions "monolayer" while there is no discussion about the potential unique role of monolayer in influencing magnetic dynamics. How do the magnetization dynamics differ from the monolayer to bulk FGT? As indicated by early experiments [e.g. Proc. Natl Acad. Sci. USA 113, E6555–E6561 (2016); Nature Photonics 13, 425–430 (2019)], The structural dynamics can be quite different in a monolayer crystal, which is expected to have a significant impact on the magnetization. Therefore, comparing monolayer and bulk results theoretically would be highly interesting to the community. Ideally, a systematic study on how the number of layers of the sample can affect the ultrafast demagnetization process should be presented. At least, the role of vdW coupling in FGT and how the layered structure impacts spin-lattice coupling should be examined and discussed.
2. The selective coupling of A_{1g} phonon with magnetization is interesting. However, the oscillation in Fig. 2c is not as periodical as the phonon modes shown in Fig. 3c. By examining Fig. 2c alone, it is difficult to justify if A_{1g} phonons are clearly present in the dynamics of magnetization. Can the authors show the FFT spectrum of the data in Fig. 2c to check if the two frequency components of A_{1g} modes are present?
3. In page 8, the discussion of spin dynamics after 800 fs gives the impression that the in-plane phonon modes are turned on after 800 fs, which seems not physical. The authors at least need to plot and examine the lattice vibration along the in-plane direction as a function of time. Why would the in-plane phonon be delayed by about 800 fs?
4. In Fig. 2c, M_z recovers strongly at 1000 fs. Does it further recover at longer time scales? Can the authors show data from 1000-1500 fs?
5. The authors claim in page 8 that "...but an increasing magnetic moment in the in-plane directions. However, Fig. 4(b) actually shows that M_x(y) decreases after 800 fs, opposite to the claim in the text.
6. The authors claim the excitation of in-plane phonon modes may be explained by nonlinear coupling with A_{1g} mode. Theoretically, it seems the calculation did not include nonlinear phononic coupling terms in Eq. (2) page 13. Experimentally, the amplitude of A_{1g} phonon excitation using optical light is too small to nonlinearly couple to other modes. Therefore, the connection between the DFT results and nonlinear phononics is quite weak. I suspect that nonlinear phononic coupling is not the underlying mechanism.
7. The coupling of the spin precession with chiral phonons is interesting and makes sense. However, the trajectories of Fe atom in Fig. 4c,d do not convincingly show well-defined circular patterns as a result of coherent chiral phonon excitation, but rather heavily damped noisy patterns.
8. Could the authors give a physical interpretation of the reduced spin-lattice coupling in nonadiabatic calculation? Is that due to strong incoherent lattice vibration or high electron temperature?

Reviewer #2:

Remarks to the Author:

The manuscript provides interesting new insights into the photoinduced demagnetization dynamics in a 2D material Fe₃GeTe₂. By selectively switching on and off spin orbit coupling, Ehrenfest dynamics, in-plane and out-of-plane phonon modes, the authors identify a three stage process: a demagnetization within the first 200fs, followed by a nearly stationary behavior for the next 600fs, related to the A_{1g} phonon mode and then another demagnetization step after 800fs which is associated with nonlinear phonon coupling. Still some questions/concerns remain that need to be clarified:

1) I encourage the authors to provide more information on the material and the contribution of the different sites: in the beginning it is stated that the material has a high Curie temperature, but no value is given and later it is said there is an antiferromagnetic to ferromagnetic transition, so can the authors clarify the ground state of the material they have considered. On p. 3 a magnetic moment of 1.53 μ_B per f.u. is given, but later in Fig 2 the value is $\sim 4.5 \mu_B$, so obviously the former value is per Fe atom. It would be important to resolve the behavior at the inequivalent Fe positions, both concerning their static behavior (e.g. spin, orbital moment, magnetocrystalline anisotropy), but also concerning the demagnetization dynamics. The role of Ge and Te is not addressed at all. How general is the observed effect and what is the origin of different demagnetization dynamics in FGT compared to e.g. elemental compounds such as Ni or Fe.

2) A major aspect is the different demagnetization behavior for what the authors describe as weak (1.71 mJ/cm²) vs. intense (1.93 mJ/cm²) excitation. The difference is however only $\sim 10-15\%$. Can the authors explain why such a small change in intensity has such a significant effect on the demagnetization dynamics, in particular beyond 600 fs.

3) In the summary the authors assert that the results provide "understanding the mechanism underlying novel phenomena related to electron-phonon-magnon coupling interactions". However, the manuscript describes mainly the coupling between spins and phonons. The coupling to the electronic degrees of freedom is not addressed, but may provide important insight. It would be helpful to clarify also which relaxation channels are considered.

4) The description of the theoretical approach is very scarce. Apart from the well-know time-dependent Schrödinger equation there is no information how the laser pulse is considered in the equations and whether linear or circularly polarized light is used. The time steps for the electronic and ionic systems differ by three orders of magnitude, how are equations of motion connected?

5) The authors state that they have considered the effect of nonadiabaticity on the magnetization dynamics for the first time. In particular they find that nonadiabatic effects reduce the spin phonon interaction. However, it is unclear how nonadiabatic effects are included to reach beyond the Born-Oppenheimer approximation common to most DFT codes. The relation between BOMD, NAMD and RT-TDDFT (with and without laser pulse?) does not become clear. Overall, the description of the methods and technical settings requires a significant improvement.

Referee #1:

The work by Wu et al. presents a theoretical investigation of the ultrafast dynamics of Fe₃GeTe₂ (FGT) using rt-TD-DFT technique. The paper is well-written and organized. Beyond the similar ultrafast demagnetization within 100 fs as previously investigated (Ref.[30]), there are very interesting observations about spin-lattice coupling from 100 fs to 1000 fs, including the selective coupling of the A_{1g} phonons to magnetism, and additional demagnetization after 800 fs. However, some of the main conclusions are not yet well supported due to the outstanding questions listed below. In addition, it is not clear if these results are uniquely related to the monolayer structure, or apply to bulk FGT, even 3D magnets. Without these discussions, the impact of the work could be limited. In summary, the manuscript has great potential but needs a major revision to meet the criteria for publishing on Nature Communications.

We thank the Referee for their overall positive feedback and for recognizing the significance of our work. We will address their comments below.

1. The title mentions “monolayer” while there is no discussion about the potential unique role of monolayer in influencing magnetic dynamics. How do the magnetization dynamics differ from the monolayer to bulk FGT? As indicated by early experiments [e.g. Proc. Natl Acad. Sci. USA 113, E6555–E6561 (2016); Nature Photonics 13, 425–430 (2019)], The structural dynamics can be quite different in a monolayer crystal, which is expected to have a significant impact on the magnetization. Therefore, comparing monolayer and bulk results theoretically would be highly interesting to the community. Ideally, a systematic study on how the number of layers of the sample can affect the ultrafast demagnetization process should be presented. At least, the role of vdW coupling in FGT and how the layered structure impacts spin-lattice coupling should be examined and discussed.

We thank the Referee for the question and for providing the references. Both Proc. Natl. Acad. Sci. 113, E6555 (2016) and Nat. Photonics 13, 425 (2019) reveal the anisotropic phonon dynamics, specifically that the in-plane phonon modes exhibit a significantly faster decay rate than the out-of-plane ones. Notably, previous investigations on FGT have also demonstrated a strong dimensionality effect on its ferromagnetism [e.g. Nat. 563, 94 (2018); Nat. Mater. 17, 778 (2018); npj 2D Mater. Appl. 4, 33 (2020)]. We agree that it is instructive to analyze the demagnetization in both monolayer and bulk FGT.

To gain a better understanding of the spin-lattice coupling strength in the layered structure, we perform new calculations to obtain the variation of the magnetic moment in the z -direction (ΔM_z) in monolayer and bulk FGT for the phonon modes of our interest. The A_{1g} phonon modes in bulk FGT are illustrated in Fig. R1, with the two layers exhibiting either in-phase or out-of-phase atomic motions. As depicted in Fig. R1, the variation of magnetic moment resulting from the same phonon displacement of the in-phase A_{1g}-1 phonon mode of bulk FGT is very similar to that obtained in the monolayer (Fig. 3 in the main text). However, the spin-phonon coupling strength for the in-phase A_{1g}-2 mode in bulk FGT is approximately one-fifth of that observed in the monolayer. This indicates that the interlayer interaction might change the electron screening in bulk metallic magnetic materials, leading to the suppression of the spin-phonon coupling compared to that in the monolayer. This observation aligns with previous work where strong spin-phonon coupling is predominately observed in fewer-layer samples [e.g. Adv. Electron. Mater. 7, 2001159 (2021)]. As for the out-of-phase A_{1g} phonons, the variation of magnetization (ΔM_z) demonstrates a quadratic dependence on the phonon displacement in bulk FGT. Moreover, the magnitude is at least one order of magnitude smaller, due to the cancellation effect between adjacent layers. Given these analyses and considering the computational cost, we think the monolayer serves as an ideal platform to study the coupled ultrafast spin-lattice dynamics and can offer insights into the fundamental physics involved. However, a full-scale TDDFT simulation on bulk and few-layer FGT is also highly desirable that we leave for future investigations.

We have made the following changes in the main text, and added the figure and discussion to the SI Sec. S3.

In Sec. The spin-lattice coupling:

“... whereas those of all other phonon modes are precisely zero. We note that the ferromagnetic behaviors in FGT have a strong dependence on its layer number [12,14,15], and the ultrafast dynamics can be quite anisotropic in low-dimensional materials [33,34]. As the ferromagnetic order is more stable in the bulk sample, the spin-lattice coupling, specifically the variation of M_z upon lattice displacement, is significantly enhanced in the monolayer system (for a detailed discussion, see SI Sec. S3).”

Figure R1: **The spin-phonon interaction in bulk FGT.** Variation of M_z for the (a) in-phase and (c) out-of-phase A_{1g} phonon modes in bulk FGT, whose atomic displacements are shown in (b) A_{1g-1} and A_{1g-2} ; (d) A_{1g-3} and A_{1g-4} , correspondingly.

2. The selective coupling of A_{1g} phonon with magnetization is interesting. However, the oscillation in Fig. 2c is not as periodical as the phonon modes shown in Fig. 3c. By examining Fig. 2c alone, it is difficult to justify if A_{1g} phonons are clearly present in the dynamics of magnetization. Can the authors show the FFT spectrum of the data in Fig. 2c to check if the two frequency components of A_{1g} modes are present?

In Fig. 3c, the projected phonon intensities highlight the predominant role of the two coherent A_{1g} modes, while in Fig. 2c, the oscillation of the magnetization moment is resulted from the contribution from all phonon modes, some of which have an amplitude approaching $0.02 \text{ \AA}\sqrt{\text{amu}}$ (in gray lines of Fig. 3c). We thank the Referee for the suggestion to check the FFT spectrum. As Fig. R2 shows, the FFT spectrum of the magnetization dynamics after 200 fs has been examined, in which there are two peaks with frequency of approximately $\omega_{\text{peak1}} = 4.72 \text{ THz}$ and $\omega_{\text{peak2}} = 8.89 \text{ THz}$. The two frequency peaks correspond to the occurrence of the two A_{1g} phonon modes, which highlights the dominant role of A_{1g} phonon modes in the magnetization dynamics. Besides, the two peak frequencies are slightly increased compared with the phonon eigenfrequency in the ground state (4.58 THz and 8.87 THz). These results are consistent with the phonon hardening effect in Fig. 5 in the main text.

We have made the following changes in the main text and added the figure and analysis to the SI.

In Sec. The three-stage ultrafast demagnetization dynamics:

“... with an oscillatory feature taking place at a timescale of a few hundred femtoseconds (yellow block on the timeline in Fig. 1). We perform a fast Fourier transform (FFT) analysis on the ΔM_z dynamics from 200 to 800 fs and observe two coherent oscillation peaks at approximately 4.72 and 8.89 THz (for details see SI Sec. S4). These two frequencies correspond to the two A_{1g} phonon modes, strongly indicating that the lattice degree of freedom plays an important role in the femtosecond magnetization dynamics which we will discuss in detail in the next session.

Figure R2: **The FFT spectrum of the magnetization oscillation during 200–800 fs.** The two dominant FFT peaks at $\omega_{\text{peak1}} = 4.72$ THz and $\omega_{\text{peak2}} = 8.89$ THz are corresponding to the frequencies of two A_{1g} phonon modes.

3. In page 8, the discussion of spin dynamics after 800 fs gives the impression that the in-plane phonon modes are turned on after 800 fs, which seems not physical. The authors at least need to plot and examine the lattice vibration along the in-plane direction as a function of time. Why would the in-plane phonon be delayed by about 800 fs?

We appreciate that the Referee raises this question. We acknowledge that the emphasis on chiral phonon dynamics after 800 fs in the text might lead to possible confusion, suggesting that the in-plane phonon modes were either absent or did not play any role until 800 fs.

In fact, as illustrated in Fig. 2c in the main text, the in-plane phonons *ALONE* did not give rise to the further demagnetization after the SOC-driven demagnetization observed in the first 200 fs. Only with the coexistence of lattice motion along both in-plane and out-of-plane directions, one can observe the lattice-induced demagnetization dynamics, where the in-plane lattice motion dominates the spin dynamics *MOSTLY*, if not *ONLY*, after 800 fs. In response to the Referee’s request for a detailed examination of the in-plane lattice vibration, we present the time-dependent projected intensities for all phonon modes in Fig. R3. Specifically, the in-plane Raman-active A_{2g} phonon modes (denoted by yellow and brown lines) are activated upon the laser excitation. During the after-pulse dynamics, two sets of doubly-degenerated infrared active phonons (in green and purple lines) are excited, with their amplitudes increasing over time, especially for the modes in purple lines after 800 fs. This is likely arising from phonon coupling with the initially activated large-amplitude Raman modes. Therefore, there exist the in-plane phonons throughout the process but the chiral ones which affect the magnetization dynamics start to play a dominant role in the magnetization dynamics after 800 fs.

Figure R3: **The time-dependent in-plane phonon dynamics.** The yellow and brown lines denote the Raman-active in-plane A_{2g} phonon modes; the green and purple lines denote two sets of doubly-degenerate infrared in-plane E_{2u} phonon modes.

We have added this analysis to the SI Sec. S5 and made the following changes in the main text.

Changed the following sentences in the third paragraph:

“... lattice vibration. Lastly, the in-plane phonon modes driven from nonlinear phononics grow in amplitude and can dominate the spin dynamics at a later stage, *i.e.* 800 fs after laser excitation. Interestingly, ...”

Added the following discussion into *Nonlinear phononics and spin precession*:

“The transverse spin relaxation evidences that in-plane phonon modes, likely driven by the anharmonic coupling with the A_{1g} modes, known as nonlinear phononics, start to significantly affect the magnetization dynamics after 800 fs. A complete presentation of the time evolution of all in-plane phonon modes can be found in SI, Sec. S5. Two doubly-degenerate infrared (IR) active modes are activated after 300 fs with their amplitudes increasing over time. Their superposition can account for the excitation of the chiral phonon modes (Fig. S9 in SI), which in turn dominate the spin dynamics after 800 fs. To better visualize such an effect, the trajectory of Fe_1 atom...”

Changed the following sentences in *Discussion*:

“... shows a clear coupling with photoexcited coherent A_{1g} phonon modes. Further, we observe that nonlinear phonon interactions between the light-driven A_{1g} modes and the in-plane phonons will introduce chiral phonon modes with an elliptical atomic trajectory, in accompany with a transverse spin relaxation, namely an effective magnetization precession after 800 fs. We anticipate that under delicate driving conditions ...”

4. In Fig. 2c, M_z recovers strongly at 1000 fs. Does it further recover at longer time scales? Can the authors show data from 1000-1500 fs?

We thank the Referee for pointing it out. Due to the substantial computational cost, we extended our simulations of the time-dependent magnetization dynamics to a longer timescale, specifically up to 1250 fs, as shown in Fig. R4. We do observe a recovery of the z -component of magnetic moment M_z after 1000 fs, as noted by the Referee. This is mostly due to the thermalization of lattice vibrations after 1000 fs, resulting in the incoherence of chiral phonons or the circular motion of Fe atoms. This is an indication that the chiral phonons driven from nonlinear phononics might not maintain a strong coherence, and their phase becomes strongly coupled with other phonon modes. If one can selectively drive the chiral phonons as proposed for example in Ref. [Nat. Phys. 13, 132 (2017)], the spin precession could be better manipulated.

We have added the corresponding figure and discussions to the SI Sec. S6, and made the following changes.

Figure R4: The time-dependent magnetization dynamics shown for the time range of 0–1250 fs. The red, green, and blue lines represent M_x , M_y , and M_z , respectively.

In Sec. *Nonlinear phononics and spin precession*:

“More importantly, although in the present case, the phonons tend to thermalize after 1000 fs (Fig. S4 in SI), which might suppress the spin precession, it is possible to optically drive the chiral phonons selectively. Once the phonon modes are coherently driven, they can maintain a better coherence, thus it is possible for a collective excitation ...”

5. The authors claim in page 8 that “...but an increasing magnetic moment in the in-plane directions. However, Fig. 4(b) actually shows that $M_{x(y)}$ decreases after 800 fs, opposite to the claim in the text.

There might be possible confusion arising from the visual representation of our plot. In Fig. 4b, the initial moments along the x and y directions are both zero. Therefore, the negative sign of $M_{x(y)}$ in the plot represents the direction of the magnetic moment, but the physical magnitude increases. Therefore there is no inconsistency in our claim.

We have made the following modification in *Nonlinear phononics and spin precession* to avoid the possible confusion:

“Therefore, we now examine both the x - and y -component of the transient magnetization (M_x and M_y) in addition to M_z , as well as the amplitude of $M_{\text{tot}} = \sqrt{M_x^2 + M_y^2 + M_z^2}$, as shown in Fig. 4(b).”

6. The authors claim the excitation of in-plane phonon modes may be explained by nonlinear coupling with $A1g$ mode. Theoretically, it seems the calculation did not include nonlinear phononic coupling terms in Eq. (2) page 13. Experimentally, the amplitude of $A1g$ phonon excitation using optical light is too small to nonlinearly couple to other modes. Therefore, the connection between the DFT results and nonlinear phononics is quite weak. I suspect that nonlinear phononic coupling is not the underlying mechanism.

We address the Referee’s question from two aspects as follows:

First, there is no explicit phonon-phonon coupling term in Eq. 2. However, within the framework of Ehrenfest molecular dynamics, the real-time atomic coordinates are evolved with the electron occupations in the excited states. Therefore, the relaxation of the electronic excitation to the lattice degree of freedom, as well as the coupling between the vibrational modes *beyond the harmonic approximation* is naturally included. The phonon intensity or amplitude is obtained by projecting the real-time lattice trajectories onto the phonon eigenmodes. Therefore, the phonon-phonon interaction or the nonlinear phononics should be well captured.

This is different from other commonly used methods to treat nonlinear phonon dynamics [*e.g.* Phys. Rev. Lett. 124, 117401 (2020); Phys. Rev. Research 3, L032046 (2021); Phys. Rev. Lett. 118, 054101 (2017)], where the coherent phonon modes are treated more or less classically based on harmonic approximations. Indeed, previous work [Phys. Rev. Lett. 131, 066401 (2023)] has shown that if the phonons are coherently driven and the decay path is dominated by a specific channel, the analytical model and molecular dynamics simulations yield similar results on the phonon-phonon coupling strength. Therefore, we are confident that our method effectively captures the nonlinear phonon dynamics on a femtosecond to picosecond timescale.

In combination with the response to Referee 2, we have expanded our method session and added Sec. S1 in the SI.

Second, the Referee raised concern on the A_{1g} amplitude that could be experimentally driven. This is a valid point. However, through optical manipulation, the pump frequency and fluence can serve as an effective knob to tune the excited phonon amplitude [Phys. Rev. Lett. 131, 066401 (2023), npj Quantum Mater. 7, 14 (2022); Phys. Rev. Lett. 128, 015702 (2022)]. Here, we did a quantitative evaluation on the dependence of A_{1g} amplitude on the pump fluence and found a quadratic increase in the phonon amplitude within the low-intensity regime, as shown in Fig. R5. In particular, it is worth noting that the distinction between the *weak* and *intense* excitation could also be seen from the amplitude of the in-plane E_{2u} phonons induced by the phonon-phonon interactions. Only when the pump fluence is beyond 1.93 mJ/cm^2 can the E_{2u} phonons reach a substantial amplitude to drive the chiral phonon and spin precession (labeled by the black and purple dotted lines in Fig. R5 to guide the eye). In combination with the second question from Referee 2, we have made a detailed comparison under a series of pump fluences, namely 1.09 and 1.71 mJ/cm^2 (weak), versus 1.93 mJ/cm^2 (intense), in terms of the phonon and magnetization dynamics, which illustrates that the nonlinear phonon interactions are indeed responsible for the observed phenomena.

Therefore, in combination with our computational framework and the direct comparison between distinct spin dynamics accompanying different phonon excitation, we attribute the nonlinear phonon coupling to be the underlying mechanism for the in-plane chiral atomic motion that drives the spin precession.

We have added an entire new section *Nonlinear phononics under different pump fluences* in the SI, and made the following modifications in Sec. *Nonlinear phononics and spin precession*:

“... as shown by the nearly-zero M_x and M_y in the entire time span. As discussed in detail in SI, Sec. S10, under weak excitation, the optically driven A_{1g} modes have much smaller amplitudes, leading to the absence of the nonlinear-phonon-induced infrared in-plane E_{2u} modes, a superposition of which can generate chiral phonons. Indeed, the amplitude of the chiral atomic motion is significantly smaller when tracing the motion of ...”

7. *The coupling of the spin precession with chiral phonons is interesting and makes sense. However, the trajectories of Fe atom in Fig. 4c,d do not convincingly show well-defined circular patterns as a result of coherent chiral phonon excitation, but rather heavily damped noisy patterns.*

We thank the Referee for their interest in our observation of chiral phonons. The present visual presentation of our data aims to explicitly display the real-time position of the Fe atom. However, as the Referee has pointed out, in the plot the pattern appears more *elliptical*, rather than forming a well-defined circle. Therefore, We here provide an alternative illustration to improve the clarity of these trajectories and to better represent the coherent chiral phonon excitation, by describing the atomic positions in terms of the azimuth angle φ , the polar angle θ , and the radical distance r in a three-dimensional polar coordinate system (Fig. R6(a)). We now plot the variation of the azimuth angle φ and the in-plane projection of the radical distance r to present the chiral phonons in the x - y plane. As shown in Fig. R6(b), each marker represents the polar coordinate (r_{xy}, φ) of the Fe trajectory, with the azimuth angle spreading from 0 to 360 degrees, consistent with the circular motion signatures. The maximum of the in-plane radius r_{xy} is 0.014 \AA , corresponding to the chiral phonon amplitude labeled in Fig. 4 in the main text. Finally, as mentioned in response to Question #4 from the Referee, the chiral phonons in this case do not exhibit an ideal coherent motion, primarily due to the presence of other phonon modes and the accompanying thermalization, which causes the noisy patterns noted by the Referee.

Figure R5: The excited A_{1g} phonon amplitudes with increasing laser fluence, with a quadratic dependence at the low intensity limit. The purple blocks show the amplitude of the infrared-active in-plane E_{2u} phonon modes excited from nonlinear phononics, showing a *threshold* behavior. The black dotted vertical line indicates the threshold for the pump fluence, while the purple dotted horizontal line schematically shows the threshold for the E_{2u} phonon amplitude to induce chiral phonons and spin precession to guide the eye.

Figure R6: (a) In a three-dimensional polar coordinate system, the position of atoms can be described in terms of the azimuth angle φ , the polar angle θ , and the radial distance r . (b) The time-dependent chiral phonon dynamics presented using the polar coordinate (r_{xy}, φ) of the Fe atom's trajectory in the xy -plane. The azimuth angle φ spreads over from 0 to 360 degrees and the maximum in-plane radius r_{xy} is 0.014 . This representation spans the time range of 0 to 1000 fs.

We have added the following comments in Sec. *Nonlinear phononics and spin precession*:

“... the maximum atomic displacement in this case to guide the eye. The Fe_1 atom indeed exhibits a chiral motion in the x - y plane, although with an elliptical trajectory rather than a well-defined circle. This might be due to that the chiral phonons driven from nonlinear phononics do not maintain a perfect coherence and the phase is strongly coupled with other phonon modes. Still, these results pinpoint that spin dynamics in the later stage are governed by a different mechanism...”

8. Could the authors give a physical interpretation of the reduced spin-lattice coupling in nonadiabatic calculation? Is that due to strong incoherent lattice vibration or high electron temperature?

This is a great question. While the precise underlying mechanism may require further investigation, our interpretation is that both the high electron temperature and nonequilibrium electron-phonon interaction contribute to this effect based on the following findings.

First, we make a direct comparison by tracking the magnetization dynamics when electron excitation is absent. Specifically, we take the transient lattice structure for each time step and calculate the magnetization, but taking the electron occupation numbers as in their *ground state*. As depicted in Fig. R7, the magnetization does not show an obvious demagnetization but rather a fluctuation near the initial M_z value. Therefore, we believe the high electron temperature, or more specifically, the nonequilibrium electron distribution will affect the spin-lattice coupling. Further, in terms of electron-phonon coupling, earlier works have also shown that the *nonadiabatic* effect, where the coupling is largely different from the perturbative picture, can drive additional electron transitions [Nano Lett. 22, 4800 (2022), Sci. Adv. 9, eadg3833 (2023)], affect superconductivity temperature [Phys. Rev. Lett. 88, 117002 (2002), Phys. Rev. B 105, 224311 (2022)], *etc.* Therefore, we believe the nonequilibrium carrier distribution plus the nonadiabatic electron-phonon coupling are possible reasons for the reduced spin-lattice coupling.

We have added in the SI, Sec. S8 to discuss the *Role of electron excitation in the spin-phonon coupling, and added the following in The nonadiabatic effects:*

“... taken into account. Meanwhile, when the electron occupation number stays unchanged from the ground state, we did not observe any demagnetization, implying that the nonequilibrium excited carrier distribution will greatly affect the spin-phonon coupling (for details see SI Sec. S7).”

Figure R7: **The role of electron excitation.** Light-driven magnetization dynamics with electron excitation (in black lines) and w/o electron excitation (in red lines).

Referee #2:

The manuscript provides interesting new insights into the photoinduced demagnetization dynamics in a 2D material Fe₃GeTe₂. By selectively switching on and off spin orbit coupling, Ehrenfest dynamics, in-plane and out-of-plane phonon modes, the authors identify a three stage process: a demagnetization within the first 200 fs, followed by a nearly stationary behavior for the next 600 fs, related to the A_{1g} phonon mode and then another demagnetization step after 800 fs which is associated with nonlinear phonon coupling. Still some questions/concerns remain that need to be clarified:

We thank the Referee for supporting our work and providing valuable suggestions. We here address their comments below.

1) I encourage the authors to provide more information on the material and the contribution of the different sites: in the beginning it is stated that the material has a high Curie temperature, but no value is given and later it is said there is an antiferromagnetic to ferromagnetic transition, so can the authors clarify the ground state of the material they have considered. On p. 3 a magnetic moment of 1.53 μ_B per f.u. is given, but later in Fig 2 the value is 4.5 μ_B , so obviously the former value is per Fe atom. It would be important to resolve the behavior at the inequivalent Fe positions, both concerning their static behavior (e.g. spin, orbital moment, magnetocrystalline anisotropy), but also concerning the demagnetization dynamics. The role of Ge and Te is not addressed at all. How general is the observed effect and what is the origin of different demagnetization dynamics in FGT compared to e.g. elemental compounds such as Ni or Fe.

We apologize for the possible confusion on the materials' properties. Since the observation of ferromagnetism in two-dimensional FGT [Nat. 563, 94 (2018)], with an estimated Curie temperature of $T_c \sim 160$ K, various studies have reported T_c within the range of 70–205 K which highly depends on the layer number [Nat. Mater. 17, 778 (2018); npj 2D Mater. Appl. 4, 33 (2020)]. With the FGT integrated with Bi₂Te₃, a T_c above room temperature is reported [ACS Nano 14, 10045 (2020)]. As to the “antiferromagnetic to ferromagnetic transition” showing up later in the introduction, we were referring to the two studies using an ultrafast laser pulse to manipulate magnetic properties in CoF₂ [Nat. Phys.16, 937 (2020)] and ErFeO₃ [Nat. Phys.13, 132 (2017)].

We have made the following changes to the Introduction to clarify this.

“Among them, a widely studied example is two-dimensional ferromagnet Fe₃GeTe₂ (abbreviated FGT hereafter), embracing a variety of advantages including **high and tunable** Curie temperature[12-15], strong electron correlation...”

“...generation of chiral phonons, and **light-induced magnetic phase transitions in systems including CoF₂ and ErFeO₃ [22,23]...**”

Regarding the static and dynamic magnetic moment in FGT, its ground state has a magnetic moment value of 5.02 μ_B /f.u., with the equal magnetic moment of Fe_I atoms each accounting for 39.2% of the total magnetic moment, while Fe_{II} contributes the rest 21.6%.

Dynamically, as we show in Fig. R8, after the laser pulse is introduced, the rapid demagnetization occurs primarily via the reduction in the magnetic moments of Fe_I atoms, while the variation of the Fe_{II} atom is less pronounced.

As for the role of the Ge and Te atoms, they did not contribute a finite magnetic moment directly with their magnetization staying at zero for both the ground and excited states (Fig. R8), yet they introduce a crucial aspect via the *lattice degree of freedom*. This stands out as one of the most important differences compared to the elemental compounds such as Ni and Fe, where the lattice mostly serves as a thermal bath, while the phonon phase space is quite limited. As we have shown in FGT, however, the optical A_{1g} phonons

induce much richer magnetization dynamics and can potentially serve as an alternative knob to manipulate spins on a picosecond, and even femtosecond ultrafast timescale.

Figure R8: **The atom-resolved magnetization dynamics of FGT.** The temporal evolution of the magnetic moment of each atom in the FGT primitive cell.

We have made the following changes in the main text, and have added the figure to SI Sec. S9.

In *The three-stage ultrafast demagnetization*:

“... with no dependence on phonons (yellow and brown lines in Fig. 2b). The demagnetization happens mostly on the two Fe_I atoms, with the Ge and Te atoms maintaining a zero magnetization during the process (for details see Fig. S7 in SI). We note that the demagnetization ...”

In *Discussion*:

“However, our results clearly demonstrate the predominant role of A_{1g} coherent phonons and nonlinear phonon interactions at different demagnetization stages. This highlights a substantial difference between such a monolayer ferromagnet and the elementary ferromagnets such as Ni and Fe, where the phonon phase space is rather limited and the lattice mostly serves as a thermal bath. Further, ...”

2) A major aspect is the different demagnetization behavior for what the authors describe as weak (1.71 mJ/cm²) vs. intense (1.93 mJ/cm²) excitation. The difference is however only ~ 10 – 15%. Can the authors explain why such a small change in intensity has such a significant effect on the demagnetization dynamics, in particular beyond 600 fs.

We thank the Referee for this comment. In the context of optical control, the materials’ responses do not necessarily exhibit a linear dependence on the pump frequency or fluence. This is partially true because the ultrafast laser pulse can drive the system *far-from-equilibrium*, where the interactions enter the nonlinear regime and the perturbative picture does not hold anymore [Phys. Rev. Lett. 128, 015702 (2022); Nat. Phys. 18, 457–461 (2022)]. This also highlights the complexity of the magnetization dynamics under non-equilibrium conditions. The coupling between the laser and the electronic degree of freedom depends on the subtle band structure, which further convolves the spin, lattice, and orbital degrees of freedom.

Specifically in our context, by naming “weak” vs. “intense”, we focus more or less on a *threshold-like* behavior based on the amplitude of optically excited phonons. In combination with response to Question #6 from Referee 1, we have plotted in Fig. R5 for the amplitude of the A_{1g} phonon modes at various fluences, showing a quadratic dependence. When the fluence increases from 1.71 to 1.97 mJ/cm², there is a signifi-

cant change in the in-plane phonon activated, shown in Fig. R9. Specifically, the chiral phonon motion can be driven by the superposition of two doubly-degenerated E_{2u} modes, which we argue are excited through nonlinear phononics. As shown in the upper and middle panels of Fig. R9(a) and (b), under the fluence of 1.71 mJ/cm^2 , the optically excited A_{1g} modes have an amplitude only a fraction of those under 1.93 mJ/cm^2 , which leads to a rather small in-plane infrared-active A_{2g} modes, and as a result an absence of E_{2u} modes. Therefore, the increase in pump fluence determines the lattice vibrations which are essential for the chiral phonon generation and the spin precession.

In combination with replying to Question #6 of Referee 1, we have added a new section discussing the Nonlinear phononics under different pump fluences and added the following discussions in Nonlinear phononics and spin precession:

“... as shown by the nearly-zero M_x and M_y in the entire time span. As discussed in detail in SI, Sec. S9, under weak excitation, the optically driven A_{1g} modes have much smaller amplitudes, leading to the absence of the nonlinear-phonon-induced infrared E_{2u} in-plane modes, a superposition of which can generate chiral phonons. Indeed, the amplitude of the chiral atomic motion is significantly smaller when tracing the motion of ...”

3) *In the summary the authors assert that the results provide “understanding the mechanism underlying novel phenomena related to electron-phonon-magnon coupling interactions”. However, the manuscript describes mainly the coupling between spins and phonons. The coupling to the electronic degrees of freedom is not addressed but may provide important insight. It would be helpful to clarify also which relaxation channels are considered.*

We thank the Referee for this question. Physically, the dynamical process we calculated includes the coupling between the electron-phonon-spin degrees of freedom. Specifically, the relaxation channels considered in our calculation include the relaxation between electronic states from the laser-induced nonequilibrium, between the electron and spin degrees of freedom through the spin-orbit coupling effect, between the electron and lattice through the electron-phonon coupling, as well as the relaxation between available phonon modes. As demonstrated, the first stage of demagnetization dynamics happens mainly via the spin-orbit coupling effect, and when the spin-orbit coupling interaction is absent, there is no significant demagnetization (Fig. 2 in the main text). Meanwhile, the electron-phonon coupling leads to the dispersive excitation of coherent phonons (DECP), which in turn affects the electronic properties and tunes the magnetic properties. Therefore, while the discussion mostly emphasizes the role of phonons in the demagnetization dynamics, which we have believed to be a major novelty of our work, our method naturally includes the relaxation of the electronic excitation to the lattice degree of freedom, as well as the coupling between vibrational modes.

Technically, 1) the Hamiltonian associated with SOC interaction can be described as $H_{\text{SOC}} = \frac{\hbar e}{4m^2c^2} \mathbf{p} \cdot (\boldsymbol{\sigma} \times \nabla V)$ [Phys. Rev. Lett. 95, 187203 (2005), npj Computational Materials 8, 145 (2022) etc.], where $V = V_{\text{ION}} + V_{\text{H}} + V_{\text{XC}} + U_{\text{ext}} + V_{\text{soc}}$ denoting the potential generated by the ions, which acts on the electrons, and \mathbf{p} represents the momentum of the electrons. In the Heisenberg picture, the kinetic velocity is $\mathbf{v} = \frac{1}{\hbar} [\mathbf{r}, H] = \frac{1}{m} \left[\mathbf{p} + \frac{e}{c} (\mathbf{A} + \mathcal{A}) \right]$ where $\mathcal{A} = \frac{\hbar}{4mce} \boldsymbol{\sigma} \times \nabla V$ which comes from the contribution of SOC. This indicates that \mathcal{A} serves as an SU(2) gauge vector potential, which can be seen as an effective magnetic field that affects the electron dynamics due to the presence of the SOC, i.e. $\mathcal{B} = \nabla \times \mathcal{A} = \hbar \left[\boldsymbol{\sigma} \cdot (\nabla^2 V) - (\boldsymbol{\sigma} \cdot \nabla) \nabla V \right] / (4mc)$. 2) The nonequilibrium electron-lattice coupling is accounted for from the Ehrenfest molecular dynamics, which we describe in detail in answering the Referee’s last question.

In combination with the response to the Referee’s following questions, we have expanded our Methods session and added the corresponding content to the SI Secs. S1 and S2.

4) *The description of the theoretical approach is very scarce. Apart from the well-know time-dependent Schrödinger equation there is no information how the laser pulse is considered in the equations and whether linear or circularly polarized light is used. The time steps for the electronic and ionic systems differ by three orders of magnitude, how are equations of motion connected?*

We appreciate the reviewer’s feedback and the opportunity to provide a more detailed description of the

Figure R9: The different behaviors of in-plane phonons under weak and intense excitation. The time-dependent phonon dynamics under laser fluence of (a) 1.09 mJ/cm², (b) 1.71 mJ/cm², and (c) 1.93 mJ/cm². The three rows, from top to bottom, correspond to the projected intensity of out-of-plane A_{1g}, in-plane A_{2g}, and in-plane E_{2u} phonon modes, respectively. The Raman-active A_{2g} are coherently driven with frequencies of 106.72 cm⁻¹, shown by the yellow curves. After 300 fs, doubly degenerate infrared-active phonon modes E_{2u}, with frequencies of 76.18 cm⁻¹ are driven due to the phonon-phonon interactions. (d) The eigenmodes of these large-amplitude phonons are visualized from the perspective along the positive direction of *a*-axis. The superposition of the doubly-degenerate E_{2u} modes can result in circular atomic motion, *i.e.* activating the chiral phonons. Under weak excitation, the E_{2u} modes have negligible amplitude thus leading to the absence of chiral phonons and spin precession.

theoretical approach used in the present work.

As to the laser pulse information, we have used a Gaussian-shaped *linearly* polarized laser pulse, whose electric field profile is shown in Fig. R10.

On the different orders of the electronic and ionic steps, it comes from the numerical treatment in the implementation of the evolution operator. Specifically, in the representation of plane wave (PW) basis sets ($\{\mathbf{G}\}$), the time-dependent Kohn-Sham states $\psi_{\gamma,\mathbf{k}}(\mathbf{G}, t)$ at each \mathbf{k} point can be expanded in the adiabatic basis sets $\{\phi_{n,\mathbf{k}}(\mathbf{G}, t_1)\}$, which are the eigenstates of Hamiltonian $H_{\mathbf{k}}(\mathbf{G}, t_1)$ [WIREs: *Compu. Mol. Sci.* 11, 1492 (2021), *Nat. Commun.* 11, 43 (2020)]:

$$|\psi_{\gamma,\mathbf{k}}(\mathbf{G}, t)\rangle = \sum_{\gamma} c_{n\gamma,\mathbf{k}}(t) |\phi_{n,\mathbf{k}}(\mathbf{G}, t_1)\rangle, \quad (\text{R1})$$

Figure R10: Waveform of the applied electric field linearly polarized along the c -axis of the crystal cell.

where the coefficient $c_{n\gamma,\mathbf{k}}(t) = \langle \phi_{n,\mathbf{k}}(\mathbf{G}, t_1) | \psi_{\gamma,\mathbf{k}}(\mathbf{G}, t) \rangle$. Writing $c_{n\gamma,\mathbf{k}}(t)$ in the form of the coefficient matrix $C_{\mathbf{k}}(t)$, the time-dependent Kohn-Sham (TDKS) equation in the adiabatic basis $\{\phi_{n,\mathbf{k}}(\mathbf{G}, t_1)\}$ is

$$H_{\mathbf{k}}(t)C_{\mathbf{k}}(t) = i\hbar \frac{\partial C_{\mathbf{k}}(t)}{\partial t}. \quad (\text{R2})$$

At an infinitesimal time interval ($\Delta t = t_2 - t_1$), the Hamiltonian in the adiabatic basis can be regarded as varying linearly with time

$$H_{\mathbf{k}}(t) = H_{\mathbf{k}}(t_1) + \frac{t - t_1}{t_2 - t_1} [H_{\mathbf{k}}(t_2) - H_{\mathbf{k}}(t_1)]. \quad (\text{R3})$$

Using the evolution operator $U_{\mathbf{k}}(t_2, t_1)$, the Eq. R2 can be written as

$$C_{\mathbf{k}}(t_2) = U_{\mathbf{k}}(t_2, t_1)C_{\mathbf{k}}(t_1) \quad (\text{R4})$$

According to Crank-Nicholson algorithm, $U_{\mathbf{k}}(t_s + dt, t_s)$ can be expanded as

$$U_{\mathbf{k}}(t_s + dt, t_s) = \exp(-i\hbar H_{\mathbf{k}}(t')dt/2) \approx \frac{1 - i\hbar H_{\mathbf{k}}(t')dt/2}{1 + i\hbar H_{\mathbf{k}}(t')dt/2}, \quad (\text{R5})$$

with $t' = t_s + dt/2$. Under the condition of sufficiently small step ($dtH \ll 1$), we interpolate N_t (in our case $N_t = 1000$) copies during the time range $\Delta t = t_2 - t_1$, i.e. $t_s = t_1 + sdt$ and $dt = \Delta t/N_t$. Now that $U_{\mathbf{k}}(t_2, t_1)$ can be expressed as:

$$U_{\mathbf{k}}(t_2, t_1) = \prod_{s=0}^{N_t-1} U_{\mathbf{k}}(t_s + dt, t_s) \approx \prod_{s=0}^{N_t-1} \frac{1 - i\hbar H_{\mathbf{k}}(t_s + dt/2) dt/2}{1 + i\hbar H_{\mathbf{k}}(t_s + dt/2) dt/2}. \quad (\text{R6})$$

From Eq. R6, the time step of the ionic system is Δt while the time step of electronic systems is $dt = \Delta t/N_t$. This is the reason why the steps for the electronic and ionic systems differ by three orders of magnitude.

To avoid possible confusion, we have expanded our theoretical framework and methods section in SI Secs. S1 and S2, and have provided more information in *Methods*.

5) The authors state that they have considered the effect of nonadiabaticity on the magnetization dynamics for the first time. In particular, they find that nonadiabatic effects reduce the spin-phonon interaction. However, it is unclear how nonadiabatic effects are included to reach beyond the Born-Oppenheimer approximation common to most DFT codes. The relation between BOMD, NAMD, and RT-TDDFT (with and without laser pulse?) does not become clear. Overall, the description of the methods and technical settings requires a significant improvement.

First, the traditional (common to most DFT codes) Born-Oppenheimer approximation assumes the electrons and nuclei are decoupled, with the nuclei considered “frozen” during the electron dynamics. The wave function for the system can be mathematically expressed as:

$$\psi(\mathbf{r}, \mathbf{R}, t) = \varphi_{\text{el}}(\mathbf{r}, \mathbf{R})\chi_{\text{nu}}(\mathbf{R}, t), \quad (\text{R7})$$

where $\varphi_{\text{el}}(\mathbf{r}, \mathbf{R})$ is the electronic wave function and $\chi_{\text{nu}}(\mathbf{R}, t)$ denotes the nuclear wave function. It is evident that, for each ionic trajectory on the potential energy surfaces (PESs), the evolution of the electron occupation number is not considered explicitly. Under nonadiabatic conditions, where the energy differences between various PESs become small, both electrons and ions have to be treated on the same footing. In theory, one has to solve the time-dependent Schrödinger equation

$$i\hbar \frac{\partial}{\partial t} \psi(\mathbf{r}, \mathbf{R}, t) = \hat{H} \psi(\mathbf{r}, \mathbf{R}, t). \quad (\text{R8})$$

According to the Runge-Gross theorem, the temporal evolution of electron wave functions is governed by the time-dependent Kohn-Sham (TDKS) equation [Phys. Rev. Lett. 52, 997 (1984)]:

$$i \frac{\partial}{\partial t} \psi_{\gamma, \mathbf{k}}(\mathbf{r}, t) = \left[\frac{1}{2m} \left(\mathbf{p} - \frac{e}{c} \mathbf{A} \right)^2 + V(\mathbf{r}, t) \right] \psi_{\gamma, \mathbf{k}}(\mathbf{r}, t), \quad (\text{R9})$$

where the velocity gauge is used and the external field appears in the kinetic term in the form of vector potential $\mathbf{A}(t)$.

The propagation of TDKS orbitals is implemented on the adiabatic basis $\phi_{n, \mathbf{k}}(\mathbf{r}, t)$

$$|\psi_{\gamma, \mathbf{k}}(\mathbf{r}, t)\rangle = \sum_n c_{\gamma n, \mathbf{k}}(t) |\phi_{n, \mathbf{k}}(\mathbf{r}, t)\rangle, \quad (\text{R10})$$

where γ and n denote the TDKS band index and the basis index, \mathbf{k} the reciprocal momentum index, and $c_{\gamma n, \mathbf{k}}(t)$ the time dependent coefficients. Expanding the wave function in the plane wave basis we arrive at Eq. **R1**.

As we have mentioned in the main text, for ions that are much heavier than electrons, their motions are treated classically on an averaged potential energy surface determined by the electronic distribution according to the Ehrenfest theorem. The nuclear positions are updated following the Hellmann-Feynman theorem [Faraday Discuss. 110, 407 (1998)]:

$$M_\alpha \frac{d^2 \mathbf{R}_\alpha}{dt^2} = - \sum_\gamma f_\gamma \langle \psi_\gamma | \nabla_{\mathbf{R}_\alpha} \left(\frac{1}{2m} \left(\mathbf{p} - \frac{e}{c} \mathbf{A} \right)^2 + V(\mathbf{R}, \mathbf{r}, t) \right) | \psi_\gamma \rangle, \quad (\text{R11})$$

where f_γ represents the occupation number of time-dependent Kohn-Sham wavefunctions. M_α and \mathbf{R}_α are the mass and position of the α th ion. Thus, Eq. (R9) and Eq. (R11) represent the *coupled electron-ion motion*. Therefore, our RT-TDDFT implementation treats the electron and nuclear dynamics on the same footing, and their coupled temporal evolution does include the nonadiabatic effect.

In the main text, the NAMD simulation is performed using the RT-TDDFT method, as described above. This involves laser excitation and considers the real-time dynamics of both electron occupation number and the transient lattice configuration. On the other hand, the BOMD simulation is initiated with *ONLY* the lattice distortion but a ground state electron occupation.

We have added relevant content in the SI and made the following changes in *The nonadiabatic effect*:

Specifically, we start from a distorted lattice structure with the two coherent A_{1g} phonon modes initiated at $t = 0$, without the optical excitation so the electron occupation number evolves from the ground state.

Reviewers' Comments:

Reviewer #1:

Remarks to the Author:

The authors have successfully addressed my concerns in the response letter and revised manuscript. This is the best response I have seen in years which is to the point and well-supported by the additional evidence. Although I was expecting a TDDFT simulation to illustrate the difference between bulk and monolayer (my point 1), I agree with the authors that it may be done separately in future work. I would recommend the manuscript as is for publication in Nature Communications.

Reviewer #2:

Remarks to the Author:

In their reply the authors have considered adequately the concerns of both referees. The clarifications, e.g. of role of the threshold fluence, the distinct demagnetization dynamics of the inequivalent Fe sites, were necessary and have helped to improve the presentation of results (very clear in the reply and mostly reflected also in the manuscript). The description of the methods part has been enhanced. One remaining question is why in Fig. R2 there are two separate zoom-ins of the two peaks instead of showing the whole spectrum. Overall, I recommend the manuscript for publication.

Referee #1:

The authors have successfully addressed my concerns in the response letter and revised manuscript. This is the best response I have seen in years which is to the point and well-supported by the additional evidence. Although I was expecting a TDDFT simulation to illustrate the difference between bulk and monolayer (my point 1), I agree with the authors that it may be done separately in future work. I would recommend the manuscript as is for publication in Nature Communications.

We thank the Referee for their valuable comments and suggestions, which have significantly helped us to improve our manuscript.

Referee #2:

In their reply the authors have considered adequately the concerns of both referees. The clarifications, e.g. of role of the threshold fluence, the distinct demagnetization dynamics of the inequivalent Fe sites, were necessary and have helped to improve the presentation of results (very clear in the reply and mostly reflected also in the manuscript). The description of the methods part has been enhanced. One remaining question is why in Fig. R2 there are two separate zoom-ins of the two peaks instead of showing the whole spectrum. Overall, I recommend the manuscript for publication.

We thank the Referee for the question. To avoid possible confusion, we here show the FFT spectrum from 3 THz to 10 THz. While there are noises at other frequencies due to numerical instabilities and the non-coherent motion of phonons present in the dynamics, the two A_{1g} modes are clearly dominant in this frequency region. We have replaced Fig. S3 in the Supplementary Information for clarity.

Figure R1: **The FFT spectrum of the magnetization oscillation during 200–800 fs.** Two dominant FFT peaks of $\omega_{\text{peak1}} = 4.72$ THz and $\omega_{\text{peak2}} = 8.89$ THz are corresponding to the two A_{1g} phonon modes.